# Fostering Environmental Awareness with Smart IoT Planters in Campuses

**DOI:** 10.3390/s20082227

**Published:** 2020-04-15

**Authors:** Bernardo Tabuenca, Vicente García-Alcántara, Carlos Gilarranz-Casado, Samuel Barrado-Aguirre

**Affiliations:** 1Departamento de Sistemas Informáticos, Universidad Politécnica de Madrid, 28031 Madrid, Spain; 2Departamento de Ingeniería Agroforestal, Universidad Politécnica de Madrid, 28040 Madrid, Spain; 3Corporación Radiotelevisión Española, Dirección de Tecnología y Sistemas, 28223 Madrid, Spain

**Keywords:** computer-based systems, environmental awareness, Industry 4.0, Internet of Things, irrigation systems, planter, project-based-learning, smart learning environments, teamwork

## Abstract

The decrease in the cost of sensors during the last years, and the arrival of the 5th generation of mobile technology will greatly benefit Internet of Things (IoT) innovation. Accordingly, the use of IoT in new agronomic practices might be a vital part for improving soil quality, optimising water usage, or improving the environment. Nonetheless, the implementation of IoT systems to foster environmental awareness in educational settings is still unexplored. This work addresses the educational need to train students on how to design complex sensor-based IoT ecosystems. Hence, a Project-Based-Learning approach is followed to explore multidisciplinary learning processes implementing IoT systems that varied in the sensors, actuators, microcontrollers, plants, soils and irrigation system they used. Three different types of planters were implemented, namely, hydroponic system, vertical garden, and rectangular planters. This work presents three key contributions that might help to improve teaching and learning processes. First, a holistic architecture describing how IoT ecosystems can be implemented in higher education settings is presented. Second, the results of an evaluation exploring teamwork performance in multidisciplinary groups is reported. Third, alternative initiatives to promote environmental awareness in educational contexts (based on the lessons learned) are suggested. The results of the evaluation show that multidisciplinary work including students from different expertise areas is highly beneficial for learning as well as on the perception of quality of the work obtained by the whole group. These conclusions rekindle the need to encourage work in multidisciplinary teams to train engineers for Industry 4.0 in Higher Education.

## 1. Introduction

The growth of wireless networks and the proliferation of mobile devices in all contexts of human social life are facilitating the implementation of initiatives in which objects and people are connected. More recently, 5G networks have increased in response speed, transfer speed, data bandwidth, and wireless coverage throughout the whole populated territory on earth [1,2]. In this context, it is key to train engineers with the capabilities to develop systems that combine people, networks, and real-world objects. The Internet of Things (IoT) is widely considered the next step towards a digital society where objects and people are interconnected and interact through communication networks in Smart Cities [3,4].

The decrease in the price of electronic components (i.e., microcontrollers, sensors, and actuators) and their availability to buy them in online markets from home has facilitated the integration of education on microelectronics in schools and universities. Manufacturers of microcontrollers, as an interested party, are putting countless “Starter kits” for sale with low-cost cases to get started in the world of IoT (e.g., Grove Starter Kit for IoT, Adafruit Microsoft Azure IoT starter kit, Arduino starter kit, SparkFun IoT starter kit, CanaKit Raspberry Pi starter kit, Vilros Raspberry Pi 4 complete). Furthermore, social networks are facilitating the sharing of the so-called “know-how” through videos and descriptive documents that help to assemble the components in only some minutes. Nonetheless, the industry not only requires engineers to know how to assemble Lego bricks correctly, but engineers must also understand why bricks are assembled this way. It is essential that universities instil ingenuity in students so that future engineers know how to apply the knowledge in the most effective, efficient and sustainable way towards Industry 4.0.

Industry 4.0 is a concept originated to describe a vision of manufacturing with all its processes interconnected through the IoT. Industry 4.0 (or 4th industrial revolution) consists of digitizing industrial processes, automating tasks by training machines with artificial intelligence, connectivity, and optimizing resources. The combination of Internet technologies and future-oriented technologies in the field of smart objects proves to be a new fundamental paradigm shift in industrial production [5]. The optimization of resources in Industry 4.0 is related to the optimization of production not only at an economic level saving raw materials and energy resources, but also at an ecological level promoting the principles of sustainability to care for the planet. The United Nations 2030 Agenda for Sustainable Development with its 17 Sustainable Development Goals presents a bold and comprehensive framework for development cooperation in the coming years [6]. In this sense, it is important to educate students about the need to implement technological solutions that meet the objectives of the 2030 agenda to ensure that the engineers of the future have more capacity and ecological awareness than their predecessors of the three previous industrial revolutions.

Implementing IoT systems that are economically and ecologically efficient requires a deep knowledge on different disciplines. Creating these systems requires expertise to interconnect the components, but also capability to implement systems that are coherent with a sustainable environment. Nowadays, engineering career Curricula are designed to train students in a specific discipline e.g., Computer Engineering, Agricultural Engineering, Forestry Engineering, Industrial Engineering, Civil Engineering. The duration of the Curricula does not usually allow training more than one discipline, and transversal competences like sustainability or reflective practice on environmental issues are not always explored enough.

In this work, we carry out a study in which future engineers from different disciplines (i.e., Computer Engineering, Agricultural Engineering) work together to implement Smart IoT planters. These Smart IoT planters aim to promote environmental awareness on university campuses. The contribution of this work is threefold: Firstly, a holistic architecture describing the Smart IoT planters implemented throughout a semester is presented. Secondly, the results of an evaluation exploring teamwork performance in multidisciplinary teams are stated. Third, alternative initiatives to promote environmental awareness on campuses from students’ perspective are presented.

This article is distributed as follows: in Section 1.1 previous work on Smart IoT planters is explored and the benefits of plants in learning spaces are specified. In Section 1.2 the scientific literature in the context of teamwork in educational settings is examined. In Section 2, the materials and methodology applied in this study are described. In Section 3 the results of the study are presented. Finally, in Section 4 the results are analysed and discussed.

### 1.1. Smart IoT Planters in Educational Contexts

Indoor plants have countless benefits to human health and well-being [7,8,9,10,11]. The classroom environment can play an important role in students’ learning and academic performance [8]. Han [9] performed a study in two different high school classes of which one served as the experimental group and the other as control. Six plants were placed at the back of the classroom. The experimental group had immediately and significantly stronger feelings of preference, comfort, and friendliness as compared to the control group. Also, the experimental group had significantly fewer hours of sick leave and punishment records due to misbehaviour than the control group. Similarly, Fjeld [10] performed a study measuring health and discomfort symptoms. The results concluded 21% lower mean score for health symptoms in classrooms with plants, and a more positive consideration of the classrooms with plants (more beautiful, brighter, and comfortable). Likewise, Khan et al. [11] explored self-reported observations on indoor air quality, aesthetics, and performance. The results show that large majorities reported that the plants improved air quality, increased pleasantness, and helped to improve their performance. This study investigates alternative implementations to install Smart IoT planters as an approach to promote environmental awareness using plants in learning spaces.

Placing IoT systems in plants is not new [12,13,14,15,16,17,18,19]. The availability of sensors and microcontrollers in the online market has facilitated the implementation of systems for monitoring plants, automate irrigation, provide artificial light, or disease detection in plants. Gomez et al. [12] designed a system for monitoring soil moisture, humidity, and ultraviolet radiation in protected crops. Srbinovska [13] presented a wireless sensor network architecture for vegetable greenhouse in order to achieve cultivation and lower management costs monitoring temperature, humidity, and illumination. Similarly, in the context of greenhouses, Lambebo and Haghani [14] designed a wireless sensor network using open source and inexpensive hardware to measure the concentration level of several greenhouse gases. These planters vary on which sensors, microcontrollers, data persistence storage, and actuators are used to grow plants. Recent implementations in higher education contexts are putting special emphasis on the technological challenges faced and on the solutions adopted [20,21], or the impact on students’ satisfaction and motivation [22]. However, none of these systems used real plants with IoT systems to foster environmental awareness in educational contexts. Hence, research questions 1 and 2 are formulated as follows:

RQ1. What kind of Smart IoT planters can be implemented to foster environmental awareness in educational contexts?

RQ2. What alternative initiatives can be proposed to promote environmental awareness in the campus?

### 1.2. Teamwork Towards Multidisciplinary Implementations

Project-Based-Learning (PBL) is a framework for teaching and learning organized activities to create a product that is becoming more popular in the last years [22,23,24,25]. Within this framework, students working in groups strive for solutions to complex problems by asking and clarifying questions, debating ideas, making predictions collecting and analysing data, communicating their findings to others and creating artefacts [26]. PBL is based on re-engineered processes that involve people from multiple disciplines to improve and broaden the competence of engineering students. The aim of PBL is to: (a) understand the role of theoretical and real-world discipline-specific knowledge in a multi-disciplinary, collaborative, practical project-centred environment; (b) recognize the relationship of the engineering enterprise to the social/economic/political context of engineering practice and the key role of this context in engineering decisions, and; (c) learn how to participate in and lead multidisciplinary teams to design and build environmentally conscious and high quality facilities faster and more economically [27]. The review of research on PBL concludes that there is a need for more research documenting the effects and effectiveness using this model [28]. In fact, the literature shows several representations to comprehend the effectiveness of learning processes in working teams [29,30,31,32,33,34], most of which have been produced around the Input-Process-Output (I-P-O) model [32]. *Inputs* set the conditions under which group interaction processes take place [35]. Inputs are variables that can affect teamwork at various levels (e.g., individual, group and environment) both internally (e.g., members’ skills, attitudes, personality, group structure, group size) and externally (e.g., level of environmental stress, reward structure). Group interaction *processes* take place when team members interact [35] and indicate how a group is performing [30]. Finally, *outcomes* are criteria to assess the effectiveness of team actions. A recent study [36] draws on the model to conclude that cooperativeness and collaborative behaviour had a positive influence on team cohesiveness, while workload and task complexity had a negative influence on it. Additionally, the study found that team cohesiveness was positively related to perceived learning, satisfaction with teamwork, and expected quality. Likewise, both perceived learning and expected quality predicted satisfaction with teamwork.

Therefore, we investigate the effects and effectiveness of PBL varying the group composition in the context of Higher Education studies [28] considering the instruments and conclusions suggested in [36]. Logically, research questions 3 and 4 are formulated as follows:

RQ3. What are the effects of multidisciplinary teamwork in learning performance?

RQ4. Which are the channels used to communicate in teamwork.

## 2. Materials and Methods

This study was carried out in the context of the Computer Based Systems module, which runs in the first semester period of the fourth year of the Degree in Computer Engineering. The module is mostly practical, and it is aimed at implementing technological solutions based on IoT for Smart Cities. This year it was decided to suggest the design of Smart IoT planters to automate the irrigation system at the university campus. Implementing an irrigation system requires very specific knowledge and expertise that most computer students do not have. For this reason, teachers decided that students of Computer Based Systems (Computer Engineering (CE)) might join students of Irrigation and Drainage Technology Systems (Agricultural Engineering (AE)) to work collaboratively to implement solutions considering multidisciplinary perspectives. Both CE and AD modules run in parallel along the first semester of the year.

As CE and AE students might broadly differ in their interests towards implementing the planter, a transversal approach was sought. Hence, teachers highlighted that the main objective of the Smart IoT planter should be to raise awareness among students and university employees about the need to cover Sustainable Development Goals (SDGs) [6] specified in modules’ syllabuses: Goal 7. Ensure access to affordable, reliable, sustainable and modern energy for all; Goal 11. Make cities and human settlements inclusive, safe, resilient, and sustainable; Goal 12. Ensure sustainable consumption and production patterns; and Goal 13. Take urgent action to combat climate change and its impacts. The modules followed a PBL approach.

### 2.1. Participants

For this study, all students enrolled (n = 40) in the Computer Based Systems module (CE), and all students enrolled (n = 12) in the Irrigation and Drainage Technology Systems (AE) were invited to participate in the study. Finally, a total of 47 students accepted the consent form and completed the pre-questionnaire (88.71% male, 14.29% female), thereof 35 were CE students and 12 were AE students. The mean age was 24.23 and the standard deviation was 4.08. All sessions were face-to-face, and the attendance was mandatory but not registered.

### 2.2. Materials

During the first month (September 2019), students were able to plan their project identifying what components they would need, how they would be assembled, and for what purpose. On the one hand, CE students took the lead within their groups by analysing the most suitable sensors, and actuators. The selection and purchase of components was restricted to the stocks in a well-known website specialized in electronic components. On the other hand, AE students took the lead within their groups to decide on which components were required to assemble the irrigation system, the type of soil, and the type of plant. Before making the full purchase, the groups presented a preliminary draft specifying which components were to be used and how they were to be assembled. Professors from both disciplines (CE and AE) validated the proposals, and suggested changes to guarantee the operation of their future IoT projects.

All students had institutional tools to work collaboratively online (i.e., Microsoft Sharepoint, Microsoft Teams, Skype, and email accounts). All students were assisted on-demand by teachers.

### 2.3. Design of the Case Study

This collaboration between faculties from different disciplines was developed with the aim to promote mixed areas of knowledge (i.e., mechatronics) within the university. This study was carried out and presented to students as an initiative in which they should design IoT systems that are compliant with the SDGs [6].

The composition of the groups varied on the expertise of the students, and the expertise of the teachers. Hence, there were 3 different types of groups:(1)Type A groups (3 x groups) comprised 4 CE students. Type A groups had support from CE teachers. These groups comprised only members from the CE discipline. These were the groups with the lowest degree of multidisciplinarity.(2)Type B groups (3 x groups) comprised 4 CE students, and 4 AE students. Type B groups had support from both the CE and the AE teachers. AE students were only assigned to type B groups. These were groups with the highest degree of multidisciplinarity.(3)Type C groups (3 x groups) comprised only CE students. Type C groups had support from both CE and AE teachers. These were the groups with a medium degree of multidisciplinarity.

Regarding the group formation, teachers presented the 3 types of groups that students might join to work along the semester. Students were assigned to groups as soon as they decided the option that was more convenient for them. Finally, there were 3 groups for each group type. Each group voluntarily chose a leader who was in charge of organizing the communication, and taking care of the materials. All groups collaborated during the semester using the channels they found handier.

CE and AE students were allocated on two different campuses in the city (i.e., Campus Sur, and Ciudad Universitaria). Hence, group leaders organized sporadic and spontaneous meetings to make progress within their projects. Additionally, students were encouraged to use the channels provided by the university to arrange synchronous and/or asynchronous online meetings.

### 2.4. Meassure Instruments

This study followed the constructs suggested by Gil et al. [36] to measure team work. Reasonably, seven-point Likert scales were used to measure the 8 constructs: (1) Cooperativeness was measured using items 4 from [37]; (2) Collaborative behaviour was measured using 5 items from [38]; (3) Task complexity and (4) Workload were measured using 3 and 4 items respectively from [39]; (5) Team cohesiveness was measured using 6 items from [37,40,41]; (6) Perceived learning was measured using items from [42]; (7) Expected quality was measured using 3 items from [43]; (8) Satisfaction with teamwork was assessed following [41,44]. The forms were completed using an online questionnaire at the end of the course.

Teachers evaluated the work of CE students in a 0 to 10 scale at the end of the course. Students had to prepare a practical demo of the planter, write a technical document describing hardware and software designs, specify how they had covered the SDGs in their implementation, suggest further initiatives to foster environmental awareness in the campus, and openly share the code in a repository. In contrast, AE students were evaluated considering elements of assessment that happened before the start of the study. Hence, AE students’ grades were excluded from the evaluation to avoid bias.

### 2.5. Procedure

This experiment took place between September 2019 and January 2020 in 32 lab sessions of two hours duration (2 sessions per week). To attract the motivation of the students, the origin of urban gardens was contextualized in World War II and linked to its current use in most developed countries where it is becoming an alternative to the consumption of transgenic foods and pesticides. Looking for a closer standpoint, this initiative was motivated in the context of their own campus to make it more sustainable.

The first two weeks of the course comprised the presentation of the module, the introduction to theoretical concepts, and the group formation. Starting the fifth school-week, each group had to deliver a preliminary draft (pre-design of the IoT system) including an outline that specified which components were going to be needed, and how they were going to be interconnected to achieve the specific objectives of their final project. The teachers used the students’ proposals to filter, to agree, and to purchase the components based on their expertise and the existing budget. Starting the sixth week, students could start working with the components. At the end of week 16th, groups defended their projects making a demo, writing technical documentation, and answering the questions formulated by the teachers during the evaluation.

### 2.6. Data Analysis

Questionnaires data and scores were imported from the survey-platform into MS Excel format and then analysed using R Studio (v1.2.1335).

The reliability coefficient (Cronbach’s alpha) was calculated to validate the internal consistency of the sample (see Table 1). Nunnally has suggested that score reliability of 0.70 or better is acceptable [45].

A Shapiro–Wilk test was conducted to confirm the normal distribution assumption of the sample towards performing an analysis of variance (ANOVA). The ANOVA test was conducted to confirm significant differences among the means obtained.

Finally, a Pearson’s correlation analysis was run to determine the relationship between the means obtained (Table 4). Pearson indicates the strength of the linear relationship between two variables for which the values range between −1 < 0 < 1. The values closer to 1 (−1) depict a stronger positive (negative) correlation, meaning that the second variable tends to increase (decrease) when the values of the first value are increased and vice versa. The closer the values are to 0, the weaker the correlation is. A p-value less than 0.01 is taken as indicator for significant correlations. We can verbally describe the strength of the correlation using the guide that Evans [42] suggested for the absolute value of r (Strength: 0.00–0.19 “very weak”; 0.20–0.39 “weak”; 0.40–0.59 “moderate”; 0.60–0.79 “strong”; 0.80–1.0 “very strong”).

## 3. Results

The results presented in Section 3.1 address RQ1, Section 3.2 addresses RQ3 and RQ4, whereas RQ2 is addressed in Section 3.3.

### 3.1. IoT Systems to Foster Environmental Awareness in the Campus

This study investigates what kind of Smart IoT planters can be implemented to foster environmental awareness in educational contexts (RQ1). Finally, the groups implemented 3 different planter types: *Vertical garden*: A wooden pallet placed vertically on a wall with 6 plastic bottles attached to it. The bottles are placed diagonally from the top to the bottom with a change of direction. When watering the bottle on top using a dripper, the water runs from one bottle to the next one immediately below. The circuit concludes in a small tank for the remaining water. See Figure 1a.*Hydroponic system*: A closed circular circuit created with pipes. The plants are placed in the holes created in the upper part of the pipe so that the base of the plant is in contact with water and nutrients. A pump is used to circulate water from a tank to the rest of the circuit. See Figure 1b.*Rectangular planters*: 6 plastic planters with 50 × 38 × 30 dimensions were installed. A slit was opened at the bottom of the side of the planters to release the remaining water. Figure 1c illustrates how some of the electronic components are embedded.

Each working group justified the selected plant considering where it was going to be installed. The groups finally decided on cyca palm, organic grass, lolium perennial, kale, artichoke, and peppermint. Ornamental flowers were planted in the vertical garden, and lettuce was planted in the hydroponic system. The composition of the soil (i.e., substratum and soil) was adapted to the needs of each type of plant.

The planters were equipped with different IoT systems. Sensors, actuators, IoT cloud platform, and irrigation systems included in the IoT systems are illustrated in Figure 2. In the following sections, a holistic approach is represented to describe all the components included by the different planters.

#### 3.1.1. Sensors

Students were able to investigate which variables they needed to manage and therefore what sensors they wanted to install in their planter. The groups selected both analog and digital sensors. Analog sensors return voltage as an output whereas digital sensors return digital values. These were the sensors installed in the planters (See Figure 2):*Water sensor*: an analog sensor which returns 0 value if no water is detected, and a higher value when water is detected. The vertical garden was designed placing the water sensor within a small tank attached to the bottleneck of the last bottle. This tank contains the excess water. Similarly, all rectangular planters have a slit in the bottom back side where excess water can escape to a plate when the pot overflows. The water sensor was placed on the plate so that whenever there was a drop on its grid, the irrigation system was automatically stopped.*Weight sensor:* an analog sensor that varies the output voltage depending on the mass on it. This sensor is placed at the base of the planter of rectangular planters to keep track of the weight. The groups installed this sensor with two different purposes: (1) Identify when to stop the irrigation. Knowing how much weight the pot has before starting to water, and how much weight the pot has when the water begins to overflow, the students were able to design a system to stop watering.; (2) Manage the evolution of the plant mass of the planter. The plant grows inside and outside the pot as time passes. This sensor allows you to know precisely what plant mass the pot has. This data is relevant to determine the amount of nutrients needed, and to determine the moment when to relocate the plant to a larger pot.*Temperature probe*: an analog sensor using a resistance that varies the output voltage between 0 and 1000 depending on the inner temperature. The probe is installed inside the land to explore the temperature of the plant at different depths.*Soil moisture sensor*: an analog sensor that measures the volumetric water content in soil. Measuring soil moisture is important to manage the irrigation system more efficiently. This sensor measures the humidity of the soil of the plant positioning the two legs inside the ground. The students proposed the use of this sensor in 2 different ways: (1) To control humidity horizontally. Students placed the sensor on the surface of the land to determine the humidity of the soil at different distances from the drip system (or plant stem); (2) to control humidity vertically. Students placed the sensor making a slit in the side of the pot at different depths to measure the evolution of the wet bulb.*Light Dependent Resistor* (*LDR*): an analog sensor whose resistance varies depending on the amount of light falling on its surface. These resistors are often used in circuits where it is required to sense the presence of light. The groups used this sensor to artificially adapt the light of the plant to make the photosynthesis, when natural light was not appropriate.*Environmental temperature*: a digital sensor that returns two decimal values reporting Celsius degrees. The temperature in campus corridors usually fluctuates depending on the time of day, the angle of the light, if windows are open or closed, number of students around, or if the heating system is on or off. Likewise, there are plants that are more sensitive than others to temperature fluctuations. The ambient temperature sensor allowed to monitor how the temperature varied throughout the day and to provide suitable feedback depending on the type of plant being cultivated.*Environmental humidity*: a digital sensor that returns two decimal values reporting percentage of humidity. Moisture is important so that photosynthesis is possible. Likewise, plants should not lose too much water from their leaves. The humidity sensor allows to monitor how the humidity varied and to adapt the feedback to the user based on the need to humidify the plant. Some students suggested installing an air humidifier as actuator in further implementations.*Environmental light*: a digital sensor, which returns four decimal values between 0 and 2400 reporting the existing light intensity measured in lux (unit of measure for the amount of light received by the sensor). Similar to the LDR, it is used to provide additional artificial light if the natural one is not enough.*CO_2_-Air quality*: a digital sensor that returns four decimal values between 450 and 2000 ppm (parts per million). Air quality is measured on the basis of the CO_2_ ppm number and volatile organic compounds coexisting in the air. The students implemented this sensor to alert users when the corridor is saturated with CO_2_ and it is necessary to open the windows.

#### 3.1.2. Actuators

*Relay*. The relay is a switch controlled by an electrical circuit by means of a coil and an electromagnet to open or close the electro valve.*Solenoid valve* (electro valve). This valve controls the passage of the irrigation water through the pipe. The valve is moved by a solenoid coil, and has only two positions: open or closed.*High intensity colour LED*. The students used sets of LEDs for two different functionalities: (1) produce artificial light to facilitate photosynthesis of the plant; (2) provide feedback to the user in real time on how the irrigation system is working. e.g., Group #1 implemented the following policy: LED lights blue when the plant is watering; the LED turns red when the AQ sensor reported over 1000 ppm of CO_2_; the LED turns green when the sensors of the plant return optimal values.*Ambient displays and feedback tools*. Students used different interfaces to show the values returned by the sensors: (a)Ambient display: The PRISMA is an environmental display to support learning scenarios [46]. See Figure 3a. The PRISMA can display information with its 24 LED ring, 8 × 8 LED matrix, and a liquid crystal display. This display was made available to students so they could configure it based on their interaction needs. E.g. Group #3 configured the PRISMA to provide a range of colours between blue and yellow derived from the humidity returned by the sensor.(b)Interactive touch screen display. This interactive display was installed to present real-time information from all planters making sensors data visible with visual metaphors. The main screen includes a menu where the user can select which planter to explore in detail. See Figure 2 (left).(c)Mobile messaging system. Group #5 configured the IoT system to send alerts by means of Telegram instant messaging app, when specific events occur. Figure 3b shows some examples of the configured alerts: “The plant has not enough light”, “Congratulations! The plant is growing under the best conditions”, “Security alert! There is no Internet connectivity. Check the planter”. The system also notifies when the irrigation has started and finished.

#### 3.1.3. Computer and Microcontroller

The board (or microcontroller) is the main component connecting the rest of the subsystems. Its role is to receive the captured data, processing the data, and send orders to actuators to maintain the plant in the best conditions. Likewise, the processor sends data to the IoT platform where it is stored and monitored according to consistent rules. The system is continuously active to periodically read, validate, and write the value of the sensors. The IoT systems implemented included both Up_2_ board (computer) and/or ESP32 (microcontroller). *UP*^2^*board*. The UP Squared board is an x86 maker board based on the Intel. The UP boards are used in IoT applications, industrial automation, or digital signage. This board is equipped with an Intel Celeron N3550 and Intel Pentium N4200 System on Chip (SoC), 40 pins, 8 GB RAM, Ethernet, HDMI, and USB connectors. This case study was carried out along the semester of the Computer Based Systems module. Hence, students were urged to implement their IoT systems using this board.*ESP32 microcontroller*. ESP32 is a series of low-cost, low-power SoC microcontrollers with integrated Wi-Fi and dual-mode Bluetooth. The ESP32 employs a Tensilica Xtensa LX6 microprocessor. ESP32 includes built-in antenna switches, power amplifier, low-noise receives amplifier, filters, and power-management modules. In this case study, the most advantageous groups were able to adapt the processing capacity of the system and replace the board with a microcontroller.

#### 3.1.4. IoT Cloud Platform

In the initial phase, students should agree on which IoT cloud platform would be used to persist and monitor data from planters. A brainstorming session was organized to explore and test existing IoT platforms. The following features were considered to take the decision: REST API, authentication type, protocols for data collection (i.e., MQTT, HTTP, CoAP), and analytics provided. The following IoT platforms were considered: Azure IoT, DeviceHive, Kaa IoT Platform, Mainflux, SiteWhere, Thingsboard.io, Thinger.io ThingSpeak, SSo2, and Zetta. Finally, Thingsboard.io received more votes from the students at the end of the brainstorming session.

All groups created a profile in Thingsboard.io to adapt the IoT platform to the specific requirements of each planter. These were the features used by the groups:*Application Programming Interface* (API). All groups used the Rest API to remotely store the data in the cloud via HTTP or MQTT protocols.*Rule engine*. Students were able to configure specific rules to validate the data and consistently perform specific actions. e.g., Group #5 configured the platform to broadcast mobile messages alerting the user via Telegram when precise events occurred.*Data persistence*. Trial profiles created in the IoT platform had restrictions regarding the duration of the data persistence in the cloud. Hence, some groups were able to configure the system to backup the data in a local database to keep long-term data.*Visualization dashboard*. The IoT dashboard is a key HMI (Human-Machine Interface) component to organize and present digital information from the physical world into a simply understood display on a computer or mobile. Hence, students were able to interpret the information stored in the IoT platform using different interfaces depending on the sensor they were able to configure (See Figure 3c).

#### 3.1.5. Irrigation System

Agricultural Engineering students and teachers were responsible for designing and installing the automatic irrigation system. The installation involved the setup of planters, motorized valves, droppers, water pipes, water counters, water filters, and pressure switches. This subsystem is formed by a relay that controls the opening and closing of a solenoid valve, which allows or not the passage of water. The relay receives a direct order from the computer/microcontroller configured by CE students. CE and AE teachers provided on-demand support but also regularly reviewed the progress performed by each group.

### 3.2. Mutlidisciplinary Teamwork on IoT

This study is aimed at exploring the effects of multidisciplinary teamwork in learning performance (RQ3). Hence, we explored the extent to which teamwork subscales can vary based on the multidisciplinarity in the composition of the work groups. The scores obtained demonstrated adequate internal consistency for 6 out of 8 scales (see Table 1). Values for Cronbach’s alpha ranged from 0.71 to 0.96 revealing sufficient score reliability for “perceived learning”, “expected quality”, “team cohesiveness”, “workload”, “satisfaction” and “collaborative behavior”. Nevertheless, values for Cronbach’s alpha ranged from 0.42 to 0.52 revealing insufficient score reliability for “cooperativeness” and “task complexity”, and they were consistently discarded for the rest of the analysis.

Means and standard deviation were calculated taking into account the composition of the groups. The results illustrated in Table 2 show that by taking together all the values of the teamwork scale in a range of 1 to 7, group B obtained an average rating slightly higher than group C. On the contrary, group A obtained a rating of 0.59 points lower than group B. Looking at the subscales individually, the results concluded that group B obtained the highest scores for “collaborative behaviour”, “satisfaction”, “team cohesiveness”, and “perceived learning”. On the other hand, group C obtained the highest scores in “expected quality” and “perceived learning”. On the contrary, group A obtained the lowest scores in all subscales.

With regard to the marks obtained by the students in the final evaluation on a scale of 0 to 10 (10 being the best score), the results showed that group B obtained the upper average grade, followed by group C. The group A scored 1.60 points lower than group B, and 1.16 points lower than group C.

The results obtained in the Shapiro–Wilk test (p-value = 0.00057) confirmed the normal distribution of the overall teamwork data sample. Exploring the teamwork subscales independently, p-values lower than 0.05 and the observations of the Q-Q plots confirm that “Collaborative behaviour”, “Satisfaction”, “Team cohesiveness”, “Expected quality”, and “Workload” samples are normally distributed. However, the p-values obtained in the “perceived learning” sample deviate from normality. Hence, “perceived learning” was consistently discarded in the ANOVA test.

An ANOVA test was performed to identify significant differences between the mean values (Table 3). On the one hand, the test resulted in significant values for the grades obtained in the evaluation (See grades in Figure 4a). On the other hand, the test resulted in Pr(>F) = 0.17 (which is slightly higher to the coefficient of significance 0.1) for the overall teamwork scales, and consequently non-significant values for the overall teamwork scales. Exploring the values obtained in the subscales, the ANOVA test resulted in significant values for “expected quality” (Figure 4b).

This study aimed at exploring the significance of the relationship between the means obtained in teamwork subscales and the grades. Additionally, we investigated the relationships within teamwork subscales (Table 4). We anticipated that learning performance (grades) would be positively correlated with teamwork subscales. Contrary to our expectations, though, the results of the analysis do not depict a significant correlation between them. Additionally, we aimed at exploring potential correlations within teamwork subscales. The results from the correlation analysis show that there is a significant very strong positive correlation between “cohesiveness” and “satisfaction”. Similarly, “cohesiveness” has a strong positive correlation with “expected quality”, “perceived learning”, and “collaboration”. Likewise, “expected quality” has a strong positive correlation with “satisfaction”.

#### Frequently Used Channels to Communicate

In this case study, we aimed at exploring the most frequently used channels to communicate while working in groups (RQ4). Students were assigned an optional task in which they could report once a week (using an online form) what channels they had used to communicate among team colleagues. There were 272 reports from 47 different students along the semester (Table 5). There was a mean of 5.78 reports by student. Whatsapp, Telegram and Teams were the most frequently used channels.

### 3.3. Educational Initiatives to Promote Environmental Awareness in the Campus

Based on the lessons learned along the semester in which the IoT planters were developed, students were encouraged to suggest alternative educational initiatives to promote environmental awareness in the campus. These were the most relevant actions:

There were different students suggesting interactive systems to improve IoT planters. Student #612 suggested developing a mobile app inspired on the Tamagotchi metaphor to take care of the IoT planters. The Tamagotchi would simulate the real IoT planter where participants could vary feeding substances or watering frequency to explore how the real planter would behave. Likewise, she suggested that the Tamagotchi might learn from the experience enabling using artificial intelligent algorithms. Following a similar approach, student #311 proposed to show a status summary of all planters in a large ambient display at the campus using Tamagotchi metaphor. Particularly, ambient displays were pinpointed as a key channel to promote environmental awareness in the campus. Student #313 proposed to show graphs illustrating water expenses by student-day, student-semester, etc., contrasting the number of litters with the ones contained in a glass of water, in a swimming pool, or the ones used by the IoT planters. Student #307 proposed to explore the operation of the planter to contrast the current energy supply with a self-supply system based on solar panels. Student #112 put forward using Twitter as feed to post actions done on the planters so anyone could track how IoT planters take care of the plants. Additionally, he also suggested posting environmental variables collected by the sensor making the data visible illustrations that catch the attention of users. These social interactions might help students understand the real environmental conditions in the campus and consequently to take action to reduce pollution in a local area. Similarly, student #311 suggested creating a hashtag (e.g., #smartIoTplanters) to track the evolutions of the plants across social networks. Student #307 proposed that voluntary students could take responsibility to take care of the IoT planters along the semester. He encouraged them to create an internal competition in which the students who were able to configure the planters with the most suited parameters according to plant, environmental conditions, but also considered introducing machine learning algorithms would save extra money for credits to study related modules at the university. Thanks to advances in technology and machine learning, chatbots have become more popular than ever in recent years. Student #912 proposed using a chatbot (text or voice chatbot) to automate remote actions regarding the planters using commands. He named it *ChatbIoT* and these would be a sample command: (1)–User: “Hi Planter 3!, can you please water the lettuces at 18:00”.–ChatbIoT: “For how long?”. User: “Five minutes please”. ChatbIoT: “Got it!. I will water the lettuces for 5 minutes at 18:00. Is that correct?”. User: “Yes, thank you Planter 3!”. ChatbIoT: “Okay, I will do so. Hasta la vista, baby.”. The ChatbIoT might also help to remember recurrent actions performed on the planter (e.g., when was the last time that the hydroponic garden was fed).

Many students suggested gamification activities to raise awareness on environmental issues in the campus. Hence, different actions and rewards were considered to implement these games: (a)Actions: Student #311 suggested that some students should be rewarded whenever they maintain the plants and take care of the IoT systems once the module finished. He also suggested creating a campaign in which students could create 1 min video pills to denounce issues happening in the campus to boost the impact in social networks. Consequently, the most impacting denounces would be rewarded. Student #111 suggested reducing paper waste rewarding students who upload their class notes into Moodle. Student #611 believed that the university could create a social game suggesting 1 individual achievable challenge for each of the of the 17 sustainable development goals. Student #612 reported that loyalty recycling in the campus should be rewarded (e.g., paper, batteries, plastics). Student #812 suggested remunerating car sharing and the use of bikes to commute to the campus.(b)Rewards: Student #111 would reward the most environmentally loyal students with scholarships, ticket bonuses (to exchange in cafeteria, coffee machines, printing machines, or events organized by the university), or grant priority to book the best learning spaces in the library (e.g., quieter, with more light). Student #114 would reward these students providing with priority to select their preferred time slots to attend to the modules distributed in alternative schedules along the day/week. Student #311 suggested providing visibility to the most active students presenting an updated ranking in visual displays.

Different students suggested organizing interactive workshops in the course of the semester. For example, Student #107 suggested creating a physical open space to show the potential benefits of using renewable energy in the campus. Student #212 proposed to create a mailbox where students could contribute with ideas to make the campus more sustainable. Student #313 argued on the importance of healthy habits, foods and plants, and considered that regular workshops should be organised to make students aware of its benefits in long term. Similarly, student #607 urged to practically explore the composition, substances, and pesticides in cultivation soils using sensors, and to promote the understanding on bio food. Student #411 recommended organizing a hackathon to develop IoT software/hardware to reward the most eco-efficient developments (i.e., computation, energy consumption).

There were different students suggesting the creation of associations to promote achievable actions to support the sustainable development goals. Student #107 suggested that creating an association would have power enough to push a “Campus without plastics “action. Student #111 believed that an association should be able to promote the replacement of toilets to save water, i.e., “a presence sensor might count the number of litters wasted every day. A monthly summary could be presented visual displays at the campus”. Student #312 recommended promoting situational awareness organizing excursions to recycling plants, or places affected by pollution. Additionally, she proposed to create a compost area in the campus garden. Last but not least, student #307 suggested the use of alternative media channels such as Radio Campus Sur to disseminate good practices in the context of environmental awareness.

## 4. Discussion and Conclusions

The rapid spread of IoT technologies has triggered the educational challenge of training future engineers to be able to design complex sensor-based ecosystems. In Industry 4.0, it is essential to educate students about the need to implement technological solutions that meet the objectives of the 2030 agenda, and to ensure that the engineers of the future consider ecological issues in their implementations.

This work presents the results of a case study in which students of Agricultural Engineer and Computer Engineer were assigned the task to create IoT systems to promote environmental awareness in the context of a university campus.

Multidisciplinary groups followed the PBL methodology to investigate IoT solutions that varied in the sensors, actuators, microcontrollers, plants, soils and irrigation system they used (RQ1). Section 3.1 elaborates on the three types of planters implemented, namely, *hydroponic system*, *vertical garden*, and *rectangular planters*. Sensors were configured considering the singularities (e.g., irrigation system) of each planter, but also reflecting on the particular care required by each plant. The holistic architecture of the components represented in Figure 2, shows that the implementation of the IoT planters covered up to 9 sensors (thereof 5 were analog, and 4 were digital). The working groups had the ability to engineer functionally different systems using the same sensors (Section 3.1.1). Different feedback systems were assembled to foster understanding on environmental issues in the campus i.e., LCD lights, ambient displays [46], mobile applications, and desktop-oriented dashboards (Section 3.1.2).

The overall architecture implemented in the case study shows that all groups were consistent featuring a three-layered architecture (Figure 5): Input layer. This layer includes the sensors collecting measurements regularly, and the processing of the data done by the microcontroller/computer (Section 3.1.3). Data is sent to the process layer via MQTT protocol.Process layer. Data is stored in a database included in the IoT cloud platform. Rules might be configured according to the specific requirements of each planter (e.g., send a mobile message via Telegram alerting the user when the *water sensor* detects spilled water). This layer includes input/output API with endpoints to store/request data in/from the IoT platform via HTTP protocol.Output layer. Actions configured to be accomplished by the actuators (i.e., alerts, enable irrigation system, disable artificial lighting system). It comprises both commands towards IoT planter maintenance (Section 3.1.5), and feedback information for third party clients using the API interface.

The architecture described in this section is frequently implemented in different engineering areas [47,48]. Therefore, IoT cloud platforms are providing improved services to facilitate a seamless integration of IoT ecosystems (Section 3.1.4). Due to the scalable nature of the proposed architecture, the ecosystem is easy to extend, and to adapt towards further implementations in educational contexts.

This study achieved the objective of fostering environmental awareness on the campus by implementing systems that addressed the following important issues: Optimization the irrigation systems using weight sensors, water sensors, and warning systems. Students implemented IoT planters that open/closed the electro-valve based on real time data, and the particular conditions required by each plant. Likewise, suitable alerts were configured to minimise energy consumption;On-site real-time feedback. The IoT planters developed included on-site feedback that draw attention to environmental and consumption variables, promoting the discussion in students walking next to the planters. The LEDs system provided visual feedback specifying when the irrigation stopped/started, when the CO_2_ level is over the configured limit, or, when the environmental sensors return optimal values for the plant. The ambient display was configured to make the soil variables visible (i.e., humidity, temperature, weight), transforming the gradient into a colour scale (Figure 3a);Online real-time feedback. The systems implemented included different software clients that obtained and reported real-time data. One example was the touchscreen display, which showed detailed graphics on the evolution of the variables of each planter (Figure 3c). Another example was the mobile chatbot, which alerts and traces irrigation schedules (Figure 3b).

Moreover, students were able to suggest alternative educational initiatives to raise awareness about environment issues in the campus (RQ2). The reported results show that the experience with the smart IoT planters facilitated students to envision multiple ingenious strategies to improve IoT planters towards educational purposes (Section 3.3). Different proposals suggested the inclusion of interactive visual and acoustic displays to increase the impact on students and employees of campus. The proposals included the use of social networks, and radio media to share the data collected by the sensors, and the creation of learning objects (in videos, text) to disseminate good practices in the use of water and energy or alert about the pollution in the campus. Furthermore, students suggested solutions based on gamification strategies, designing ingenious reward policies for active users raising environmental awareness. In this context, students were able to identify different achievable activities that could be promoted within any campus. These initiatives imply an important base of knowledge towards implementing further actions to promote environmental awareness in educational contexts.

Multidisciplinary education is key to tackle complex projects covering different areas of knowledge. The implementation of smart IoT planters in educational contexts demanded technical expertise to assemble hardware components and programing software interfaces. Nonetheless, it also required specific expertise on agronomics to select convenient plants, soils, planters, and, to install suitable irrigation systems. In this case study, we wanted to investigate how multidisciplinarity working in groups might impact learning performance and the expected quality of the outcomes (RQ3). The results presented in Section 3.2 show that multidisciplinary work among students from different areas of knowledge is ostensibly beneficial for learning. The groups with the highest degree of multidisciplinarity obtained higher *grades* (Figure 4a). Likewise, a similar effect was observed for the perception of *expected quality* of the work (Figure 4b). Exploring the subscales comprising teamwork, the results of the analysis show that the perception of group cohesion (*team cohesiveness*) and *satisfaction* to work in groups are strongly correlated (Table 4). These findings are consistent with previous research concluding that *expected quality* predicts satisfaction with teamwork [36]. These results rekindle the need to promote work in multidisciplinary groups from different areas of expertise to achieve a deeper knowledge, and to create functionally efficient IoT systems in Higher Education contexts.

Work-in-groups implied both face-to-face and remote to collaboration among group members. Hence, students could freely use different channels of communication to coordinate their activities. On the one hand, they had a default institutional platform (based on Microsoft Teams), which featured synchronous and asynchronous messaging system. Alternatively, they might use any personal messaging applications or social networks. The results collected during the course show that students were more active using personal messaging application from their own mobile devices (Table 5): i.e., WhatsApp, Telegram. However, students were reluctant to use social networks for academic purposes: i.e., Facebook, Instagram (RQ4). These results must be interpreted together with recent conclusions on mobile learning showing that the new generation of teachers would be willing to use personal devices to guide students in educational contexts [49].

The complexity of this study lied in the implementation of a PBL activity across a full semester synchronizing students from two different engineering degrees (Computer Engineering and Agricultural Engineering), campuses, and modules (Computer based systems, and Irrigation and Drainage Technology Systems). This approach is especially difficult to implement since engineering degree syllabi are usually tightly restricted to specific areas of expertise. The results reported in this work represent an important technical knowledge base towards the implementation of IoT ecosystems in CE and AE educational contexts. Likewise, the conclusions of this evaluation provide evidence of the need to encourage work in multidisciplinary teams to train engineers towards Industry 4.0.

Further research should explore alternative associations between multidisciplinary groups to define suitable IoT architectures towards suggesting Appendix A learning/teaching paths for industrial, civil, naval, aerospace, or forestry engineering studies.

## Figures and Tables

**Figure 1 sensors-20-02227-f001:**
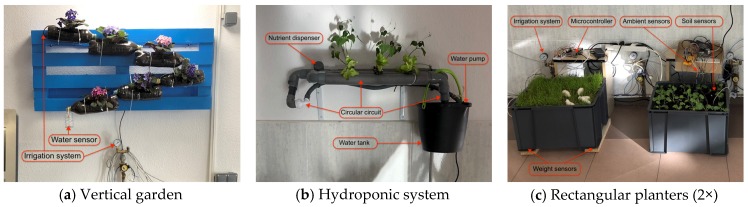
Internet of Things (IoT) planters installed in the campus.

**Figure 2 sensors-20-02227-f002:**
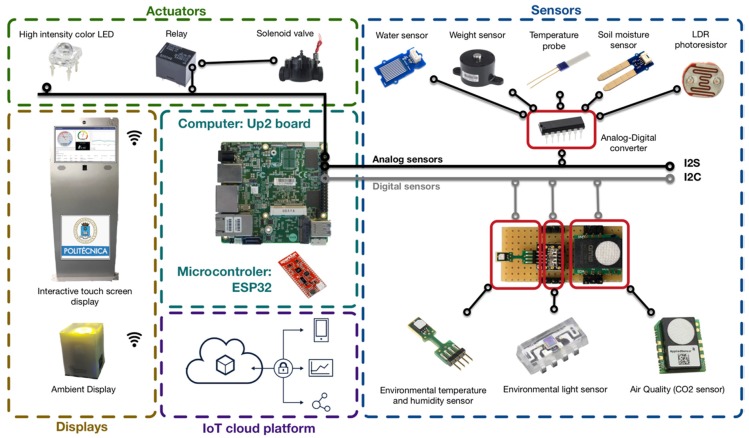
Holistic architecture of the components included in the IoT planters.

**Figure 3 sensors-20-02227-f003:**
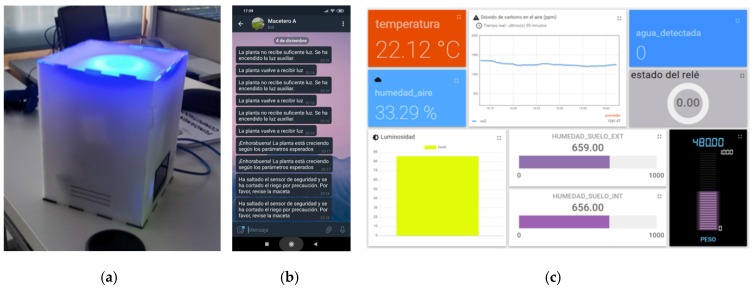
Feedback services configured: (**a**) Prisma, a visual feedback display; (**b**) Telegram messenger to receive alerts; (**c**) IoT cloud platform. Thingsboard.io desktop dashboard.

**Figure 4 sensors-20-02227-f004:**
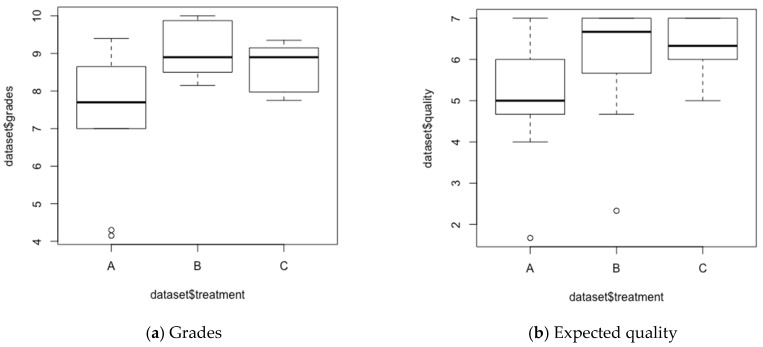
Boxplots contrasting group composition.

**Figure 5 sensors-20-02227-f005:**
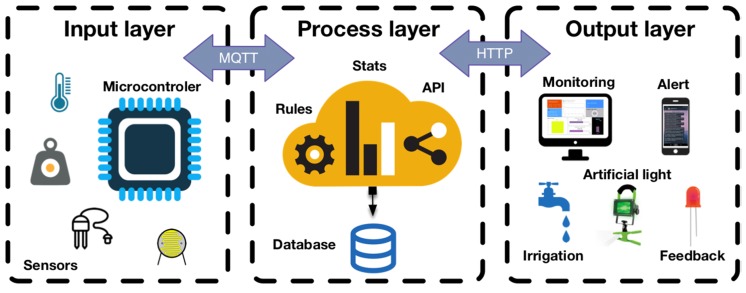
Frequently implemented three-layered architecture in IoT ecosystems.

**Table 1 sensors-20-02227-t001:** Overall Means (M), Standard Deviations (SD), and Reliability Coefficient (Cronbach’s Alpha).

Scale (7-Point Likert)	M(SD)	Cronbach’s α	Sample Item
Perceived learning	4.35(1.68)	0.96	*I am learning to identify the central issues of the subject*
Expected quality	5.75(1.30)	0.94	*I think the work of my team deserves a high mark*
Team cohesiveness	5.67(1.23)	0.89	*Every member in the team fulfils their part*
Workload	2.78(1.38)	0.89	*Teamwork requires a lot of my time*
Satisfaction	5.97(1.08)	0.77	*I enjoy working with my team*
Collaborative behaviour	5.70(0.94)	0.71	*Teamwork is stimulating for me*
Cooperativeness	5.06(0.78)	0.52 *	*I like to work with other people*
Task complexity	4.27(1.25)	0.42 *	*I have undertaken similar tasks in other subjects*

*: Internal consistency (α ≥ 70).

**Table 2 sensors-20-02227-t002:** Teamwork means and standard deviations by group composition.

Scales/Subscales	Group A	Group B	Group C
	M(SD) N = 15	M(SD) N = 12	M(SD) N = 12
**Overall teamwork**	4.67(0.88)	5.26(1.01)	5.24(0.61)
Collaborative behaviour	5.69(0.91)	5.80(1.10)	5.58(0.85)
Satisfaction	5.55(1.32)	6.47(0.89)	5.95(0.76)
Team cohesiveness	5.28(1.33)	5.93(1.28)	5.84(1.04)
Expected quality	5.12(1.32)	5.97(1.43)	6.27(0.75)
Perceived learning	4.00(1.63)	4.65(1.79)	4.47(1.71)
Workload	2.38(0.69)	2.72(1.63)	3.31(1.63)
**Grades**	7.49(1.63)	9.09(0.69)	8.65(0.63)

**Table 3 sensors-20-02227-t003:** Analysis of Variance ANOVA. Significance Pr(>F) > 0.1.

Scales	Sum of Squares	df	Mean Square	F	Pr(>F)
**Teamwork**	**2.27**	**2**	**1.35**	**1.84**	**0.17**
Collaborative behaviour	0.26	2	0.13	0.14	0.86
Workload	5.25	2	2.62	1.41	0.25
Team cohesiveness	3.11	2	1.55	1.02	0.37
Expected quality	8.56	2	4.28	2.81	0.07
Satisfaction	5.04	2	2.52	2.31	0.11
**Grades**	**17.14**	**2**	**8.57**	**6.73**	**0.003 ****

Pr(<F) Significance codes: 0 ‘***’ 0.001 ‘**’ 0.01 ‘*’ 0.05 ‘.’ 0.1 ‘ ’ 1.

**Table 4 sensors-20-02227-t004:** Pearson’s correlation analysis (* Correlation significance < 0.01).

r	Grades	Collaborat.	Workload	Cohesive.	Learning	Quality	Satisfact.
Grades	1						
Collaboration	−0.03	1					
Workload	0.09	−0.12	1				
Cohesiveness	0.09	**0.61 ***	0.12	1			
Learning	0.09	**0.59 ***	−0.01	**0.62 ***	1		
Quality	**0.34**	**0.40**	0.12	**0.66 ***	**0.35**	1	
Satisfaction	**0.31**	**0.56 ***	−0.02	**0.81 ***	**0.47**	**0.67 ***	1

**Table 5 sensors-20-02227-t005:** Channels used to communicate among team colleagues. Frequency of usage.

	Never 1	Rarely 2	Occasionally 3	Frequently 4	Very Frequently 5	M(SD)
	%(n)	%(n)	%(n)	%(n)	%(n)	
**WhatsApp**	14.34 (39)	5.51 (15)	12.50(34)	22.43(61)	45.22(123)	3.79(1.43)
**Telegram**	47.06 (128)	7.35 (20)	5.15(14)	5.15(14)	35.29(96)	2.74(1.83)
**eMail**	53.68 (146)	20.59 (56)	15.81(43)	6.62(18)	3.31(9)	1.85(1.11)
**Teams**	74.63 (203)	13.97 (38)	9.19(25)	2.21(6)	0(0)	1.39(0.75)
**Skype**	87.13 (237)	9.56 (26)	2.21(6)	0.74(2)	0.37(1)	1.18(0.53)
**Facebook**	99.26 (270)	0.74 (2)	0(0)	0(0)	0(0)	1.01(0.08)
**Instagram**	99.26 (270)	0.74 (2)	0(0)	0(0)	0(0)	1.01(0.08)

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
