# Peer review of "Fostering Environmental Awareness with Smart IoT Planters in Campuses"

_sensors, 2020, doi:10.3390/s20082227_

Round 1
Reviewer 1 Report
The manuscript presents an educational oriented paper where using problem-based learning, two groups of students develop an IoT powered planter system in order to foster environmental awareness in the educational settings.
The study presents solid and methodological use of metrics to support their findings.
However, the scope of the paper has some issues:
1.- In the abstract section, It is not clear if the paper is educational driven or solve/addresses concrete educational issues.
2.- The main result is highly questionable in engineering settings: There is a lot of evidence that concludes the same that the result of the manuscript; as authors establish: “The results of the evaluation provide evidence that multidisciplinary work including students from different areas of expertise is ostensibly beneficial for learning as well as on the perception of the quality of the work obtained by the whole group”.
It is well known that working in groups of diverse disciplines brings enormous benefits to the goal of a project as the authors conclude. The synergy and interdisciplinarity of groups empower the learning curve of individuals and improves significative the result of any engineering goal.
3.- The contribution of architecture it is not innovative from an engineering point of view: The so-called "Three-layered architecture" is common in engineering cause it only an input --> Process --> output processing type block (fig. 5).
The findings of the paper may be of higher interest into a pedagogical/educational Journal.
Reviewer 2 Report
The study addressed an area that lacked rigorous research and investigation but of great importance. The authors utilised the Project-based Learning framework to evaluate how a group of people can come up with a solution to a complex problem (in this case environment-aware planters, smart agriculture and the Industry 4.0 is the context) in the presence of various IoT devices and other electronic components such as microcontrollers. Overall, it's a thoughtful study, and its texts are in good shape. Nevertheless, the following provides specific comments that this reviewer think authors can take into account to improve the quality of this work further.
Page 177
Authors didn't justify why they selected 40 Computer Engineering (CE) students and only 12 Agricultural Engineering (AE) students. They must have a rationale behind this selection that I think should be explained.
Page 181
Both CE and AE students received with a set of devices to pick from. Again I find no rationale is given. I strongly recommend authors to provide justifications as to why these devices are made available to the respective group of students.
Page 188
The design of the case study is an essential part of this paper. It certainly deserves to be presented more thoroughly. I recommend the authors to expand this section providing more detailings.
Page 227
Data analysis subsection is too small. I expected authors to explain the methods and their justifications. Later I found the authors provided detailed analysis with results. However, it seems odd to read a lot of results without knowing their suitability in this study.
Page 446
There are statements such as, ``A Pearson's correlation analysis was run to determine the relationship between the means obtained in teamwork subscales and the grades.''. I, however, didn't find why these findings are important to prove the research questions set out earlier. There are many such instances in this paper. I don't want to argue that these are not important, but the authors must make a connection to tie-up the story.
Page 551, 561, 586
In the discussion section and subsequent explanations, the authors tried to prove the research questions. I wouldn't say they failed in their attempt but found a lack of connections between their results and the discussions. It's better if they refer back to specific findings from the analysis to establish the facts required to prove the research questions.
In a nutshell, it's an excellent study described loosely. The authors have the room to improve it further by providing justifications of their choices and tightening up the arguments referring back to the statistical analysis.
Reviewer 3 Report
This paper has reported a holistic architecture describing the IoT planters. Then the teamwork performance in multidisciplinary teams is evaluated. Data analysis is conducted to supporting the conclusions that the authors drawn. However, there are still some problems that author should explain or revise.
- In section 1, there are few references about the latest research. It is recommended to cite the latest articles [RefA-B] on the IoT field.
“5G NFV-Based Tactile Internet for Mission-Critical IoT Services,” IEEE Internet of Things, publication online, DOI: 10.1109/JIOT.2019.2958063
- Please explain what the abbreviations mean when they first appear, such as SDGs on page 4.
- Why did you choose Goal 7,12,13,14 to achieve? How important are these four goals compared to the remaining 13 goals?
- Of the 47 participants, how many are AE and CE students?
- In case study design, there are three different types of groups. In type C groups, why use informatics students instead of AE students?
- In section 2.4, the score for AE students was evaluated using a different criterion. Hence, AE students were excluded from the results reported in this case study. Please explain the reason for using the different evaluation criteria.
- In Figure 1, each part of the IoT planters is not clearly indicated. Please indicate the positions of the sensors, actuators and IoT cloud platform in Figure 1.
- In section 3.1.5, the author only mentioned the responsibilities of CE students and teachers, without explaining what the AE students and teachers are responsible for.
- As the authors mentioned that one of the contributions is to promote environmental awareness on campuses from students’ perspective. However, the promotion of environmental awareness is not reflected in the results. Please add the corresponding result to support your opinion.
Round 2
Reviewer 1 Report
The manuscript is in good shape. Authors have addressed and responded all comments and suggestions made by this reviewer